# Characterization of a Topically Testable Model of Burn Injury on Human Skin Explants

**DOI:** 10.3390/ijms21186956

**Published:** 2020-09-22

**Authors:** Olivia Gross-Amat, Marine Guillen, Damien Salmon, Serge Nataf, Céline Auxenfans

**Affiliations:** 1Lyon-Est School of Medicine, University Claude Bernard Lyon-1, 69100 Villeurbanne, France; marine.guillen@univ-lyon1.fr (M.G.); serge.nataf@inserm.fr (S.N.); 2Bank of Tissues and Cells, Lyon University Hospital (Hospices Civils de Lyon), 69003 Lyon, France; celine.auxenfans@chu-lyon.fr; 3CarMeN Laboratory, INSERM U1060, INRA U1397, INSA de Lyon, 69600 Oullins, France; 4Fundamental, Clinical and Therapeutic Aspects of Skin Barrier Function, EA4169, University of Lyon 1, 69008 Lyon, France; salmondam@hotmail.fr; 5Tissue Biology and Therapeutic Engineering Laboratory, UMR 5305, 69007 Lyon, France

**Keywords:** burn, human skin, ex vivo, wound healing, poloxamer hydrogel

## Abstract

Severe burn injuries remain a major health problem due to high rates of mortality, residual morbidity, and/or aesthetic damages. To find new therapies aimed at promoting a harmonious healing of skin burns, it is important to develop models which take into account the unique properties of the human skin. Based on previously described models of burn injury performed on human skin explants, we hypothesized that maintaining explants under constant tension forces would allow to more closely reproduce the pathophysiological processes of skin remodeling. We thus. Here, we set up and characterized an improved model of deep second-degree burn injury on ex vivo cultured human skin explants at air-liquid interface and maintained under conditions of constant tension forces. A spontaneous re-epithelialization of the lesion was observed 8 to 9 days post burn and was found to rely on the proliferation of basal keratinocytes at the wound edges. Collagen VII at the dermo-epidermal junction reformed along with the progression of re-epithelializatio and a synthesis of procollagen III was observed in the dermis at the wound site. These findings indicate that our model is suitable for the assessment of clinically-relevant therapies aimed at modulating the kinetics of re-epithelialization and/or the activation of fibroblasts following skin burn injuries. In this regard, we evaluated the use of a thermoreversible poloxamer hydrogel as a vehicle for topically-testable therapeutic molecules. Our data showed that, although useful for drug formulation, the p407/p188 poloxamer hydrogel induces a delay of skin re-epithelialization in humans skin explants submitted to experimental burn injury.

## 1. Introduction

Burns, irrespective of their severity level, are one of the most common causes of accidental injuries worldwide [1,2,3,4]. In first-degree burns, only the superficial layers of the epidermis are affected, allowing tissue repair via the mobilization of the keratinocyte stem cells and progenitors residing in the basal layer [5]. As a result, first-degree burns heal spontaneously without scar formation and do not require any treatment other than a topical moisturizer [6]. In contrast, third-degree burns do not heal spontaneously and require surgical intervention to cover wounds (autografts, allografts, epidermal sheets) [5,7]. Although current treatments have greatly increased the survival rate of patients suffering from extended third-degree burns [8], it is estimated that burns cause 180,000 deaths per year according to a recent report of the World Health Organization. In addition to being potentially fatal, severe skin burns may result in long-term functional and/or aesthetic complications [9], notably when the healing process is hampered by secondary infections and abnormal tissue remodeling. [3]. Finally, superficial partial-thickness (superficial second-degree) burns spontaneously heal within a time period of 7 to 14 days post-burn, as there are still mitotically active stem cells in the basal layer. On the other hand, deep second-degree burns destroy the entire epidermis, including the cells of the basal layer, and result in the formation of a scar. Delayed complications may occur during the healing process of deep second-degree burns requiring then the use of surgical treatments [6,7]. Currently, the most commonly used treatment for second-degree burns worldwide consists of the topical application of 1% silver sulfadiazine. However, despite the effectiveness and ease of use of such a topical treatment, side effects of silver sulfadiazine have been reported such as hyperpigmentation, delayed wound closure, and neutropenia [10,11]. It is, thus, essential to improve the management and treatment of second-degree burns and to generate a reliable and standardized model performed on human cells. Indeed, while a number of potential molecular targets have been identified in animal models of deep second-degree burns, the clinical translatability of such findings needs to be confirmed by ex vivo or in vivo human models. In particular, it is now well established that the human skin exhibits unique features in terms of histological structure and ability to heal via re-epithelialization rather than skin contraction as observed in rodents [12,13,14]. The skin of pigs harbors similarities with the human skin; however, issues linked to ethics, reproducibility, and cost [15] render difficult the setting up and use of burn models performed in pigs. Alternatively, human skin explants have been successfully used to mimic some aspects of skin burn injuries [12,16,17,18]. However, to our knowledge, such experimental models did not take into account the impact of tension forces on skin remodeling. In this context, our objective was to develop and characterize an improved model of second-degree burn injury performed on ex vivo cultured human skin explants cultured at air-liquid interface and maintained under constant tension forces. On this model, we performed a kinetics analysis of (i) re-epithelialization, (ii) keratinocyte proliferation, (iii) repair of the dermo-epidermal junctions, and (iv) tissue remodeling of the underlying dermis. Importantly, we tested the use of a thermoreversible poloxamer hydrogel as vehicle for the future evaluation of topically applied drugs aimed at treating second-degree burns. We discuss the advantages and possible drawbacks of our model as compared to previously published experimental models of second-degree burns.

## 2. Results

### 2.1. Histological Characterization of Ex Vivo Cultured Human Skin Explants

In order to optimize our protocol of ex vivo cultured skin explants grown at the air–liquid interface, we compared two different culture media (Figure 1) with regard to the general skin histological structure. To this aim, hematoxylin phloxine saffron (HPS) staining was performed on skin explants on days 0, 7, 14, and 21 after culture initiation. When comparing results obtained with medium A (a standard culture medium) vs. medium B (a culture medium routinely used for three-dimensional (3D) cultures of human skin cells) (see detailed composition in Section 4), we observed that, at late time points (days 14 and 21), the epidermis tended to detach from the underlying dermis in skin explants cultured in medium B. Although medium B was a priori more suited to our experimental settings, we opted for medium A. Under these experimental conditions, the histological structure of skin explants remained overall unchanged. However, a decrease of 40% in nucleated epidermal cells could be demonstrated on day 14 (data not shown). On the other hand, the thickness of the stratum corneum, a skin layer exclusively formed by non-nucleated cells, increased progressively from day 0 to day 14, a process likely due to the lack of mechanical friction forces exerted on skin explants. In order to evaluate the dynamics of epidermal cells renewal in cultured skin explants, Ki67, an antigen present on a protein expressed at the time of proliferation, localized to the chromosomes in mitosis [19], commonly used as a marker of proliferation, was evaluated at different time points following culture initiation (Figure 2a). As expected, the observed decrease in nucleated epidermal cells in cultured skin explants was accompanied by a parallel decrease in keratinocyte proliferation index (Figure 2a). While skin explants on day 0 contained on average 6% proliferating keratinocytes, there were still 4% proliferating keratinocytes on day 5, but this percentage dropped to 1–2% on days 9 and 14 (Figure 2b).

### 2.2. Setting up of an Experimental Model of a Deep Second-Degree Burn Injury on Ex Vivo Cultured Human Skin Explants

A metal rod heated to 100 °C was applied for different time periods (1 s, 5 s, and 10 s) on ex vivo cultured human skin explants (Figure 3). HPS staining was then performed on burnt areas at 10 min post burn, and on days 7, 14, and 21. We observed that a 1 s long application impacted only the superficial epidermal layers without damaging the basal layer (days 0 and 7) and was followed by a full re-epithelialization (day 14), which corresponds to a model of first-degree burn. In contrast, a 5 s or 10 s long application of the heated metal rod induced lesions of the whole epidermis layers as assessed by HPS staining. More specifically, we found that, 10 min after experimental burns were performed, epidermal cells were vacuolized throughout all epidermal layers and the epidermis was detached from the dermis. On day 7, following the debridement step performed on day 1, the burnt area was devoid of epidermis, which corresponds to the histological features of a deep second-degree burn. Following either 5 s or 10 s long experimental burns, the epidermis was fully re-epithelialized on day 14 post burn. This re-epithelialized epidermis comprised all the epidermal layers up to the stratum corneum (Figure 3). Since burns of longer durations are more likely to also impact the dermis, we chose to further assess our model under conditions of 10 s long experimental burns.

### 2.3. Kinetics of Re-Epithelialization in Human Skin Explants Subjected to Experimental Deep-Second Degree Burn Injury

To investigate the kinetics of re-epithelialization in our model of experimental burn injury, serial histological analyses were performed on lesioned skin explant samples obtained from three distinct donors. The extent of re-epithelialization was assessed on days 0, 1, 3, 5, 6, 7, 8, 9, 10, and 14 post burn by HPS staining. Slight interindividual differences were observed regarding the time point from which re-epithelialization initiates (day 8 for two donors, day 9 for one donor). Numeration of Ki67^+^ proliferating keratinocytes was obtained on days 0, 5, 8, 9, and 14. However, the same series of events took place in the burnt area, as exemplified in Figure 4: (i) proliferating keratinocytes formed a monolayer of cells on a restricted area localized on the edges of the lesion (re-epithelialization, day 1); (ii) the whole lesioned area was then progressively colonized by a monolayer of proliferating keratinocytes (re-epithelialization, days 2–4); (iii) the epidermis progressively thickened from the edges to the center of lesions, leading eventually to the formation of a stratum corneum layer (re-epithelialization, day 6). Of note, before re-epithelialization of the injured skin initiated, the percentage of proliferating keratinocytes increased on day 5 post burn in the non-lesioned areas adjacent to the lesion edges (Figure 4c). Interestingly, when a hair follicle could be observed in such adjacent non-lesioned areas, we could also demonstrate the presence of Ki67 proliferating cells which appeared to form a migratory stream from hair follicles toward the lesion edges (Figure 4d). Initiation of re-epithelialization was paralleled by a marked increase in keratinocyte proliferation index which reached 56% at the wound margins (Figure 4b,c). Such an index then decreased over time along with the progression of re-epithelialization from the edges to center of lesions and in parallel with the progressive thickening of the epidermis. Thus, on day 14 post burn, the keratinocyte proliferation index reached similar levels (10%) to that observed in non-lesioned skin explants on day 0 of the culture protocol (6%).

### 2.4. Kinetics of Dermal Remodeling in Human Skin Explants Subjected to Experimental Deep Second-Degree Burn Injury

In order to quantitatively and qualitatively assess dermal remodeling in our experimental model of deep second-degree burn injury, we performed serial immunostainings of collagen I, collagen III, procollagen III, and α-smooth muscle actin (αSMA), a marker of myofibroblasts, on skin explants on days 0, 5, 8, 10, and 14 post burn. While the levels and expression patterns of collagen I, collagen III, and αSMA appeared unchanged at the different time points studied (Appendix A), we observed a marked decrease in procollagen III staining intensity starting on day 8 in the dermis underlying the burnt epidermal area (Figure 5). Although less pronounced, such a decreased staining intensity was still observed on day 10 post burn. On day 14, procollagen III staining intensity returned to basal levels and was comparable to day 0 staining. It should be noticed that, on day 5 post burn, we repeatedly found that human leukocyte antigen (HLA) class II positive cells appeared to accumulate in the wound margins in both the epidermis and dermis (Appendix A). This observation, which may reflect a nonspecific activation of immune or nonimmune cells in particular blood vessels (Appendix A), indicates that human skin explants are endowed with a certain level of immunocompetence, which may prove to be of interest in future studies.

### 2.5. Kinetics of Dermo-Epidermal Junction (DEJ) Restoration in Human Skin Explants Subjected to Experimental Deep Second-Degree Burn Injury

To assess the kinetics of DEJ regeneration in our model of experimental deep second-degree burn injury, a staining of collagen VII was carried out on days 5, 8, 10, and 14 post burn (Figure 6). Interestingly, while no collagen VII staining remained present at the DEJ on day 5 post burn, the process of re-epithelialization was temporally associated with a progressive restoration of the DEJ. Such a process could be demonstrated at the DEJ of newly formed monolayers of keratinocytes, in the lesion edges. However, the intensity of collagen VII staining appeared to also parallel the progressive increase in epidermis thickness, which similarly started in the lesion edges before extending toward the center of the burnt area (Figure 6a,b).

### 2.6. The Topical Application of a Poloxamer Hydrogel Delays but Does Not Prevent Tissue Repair in Human Skin Explants Subjected to Experimental Deep Second-Degree Burn Injury

We then sought to determine whether our experimental model could be suitable for the therapeutic assessment of topically applied regenerative compounds. To this goal, we assessed the impact of a poloxamer hydrogel which was previously shown to exhibit exploitable vehicle properties in the context of pharmacological tests [20,21,22]. Daily topical applications of such a poloxamer hydrogel were performed on areas of lesioned skin explants in our model of experimental second-degree burn injury. Serial HPS staining and immunolabeling of Ki67^+^ proliferating keratinocytes were then performed of skin explants on days 5, 8/9, 10, and 14 post burn (Figure 7b). Surprisingly, we constantly observed that initiation of re-epithelialization was delayed in hydrogel-treated skin explants as could be observed only on day 10 as compared to day 8 in untreated burnt explants (Figure 7a). However, a full re-epithelialization was achieved on day 14 in hydrogel-treated skin explants, indicating that, once re-epithelialization is initiated, tissue repair is otherwise unchanged by the topical application of a poloxamer hydrogel. Favoring this view, the proliferation index of keratinocytes in the lesioned area was delayed by roughly 2 days but followed a kinetics which was strictly similar to that observed in untreated skin explants submitted to experimental injury (Figure 7c). Thus, on day 5, 7% proliferating keratinocytes were observed at the edges of the gel-treated burn areas compared to 27% for untreated burns (Figure 7c). In the hydrogel-treated burn area, the percentage of proliferating cells reached 47% on day 8, which is similar to the proliferation index observed in untreated burnt explants (55%). On the other hand, on day 14, the hydrogel-treated burn still contained 33% proliferating keratinocytes compared to 10% in untreated burnt explants.

Topical treatment with poloxamer gel, thus, causes a delay in re-epithelialization and a delay in keratinocyte proliferation but does not prevent or qualitatively modify the process of epidermal tissue repair.

We also assessed the impact of poloxamer hydrogel on the dermal extracellular matrix remodeling and focused our analysis of the kinetics of procollagen III staining alterations (Figure 8). As observed with regard to re-epithelialization and keratinocyte proliferation index, we found that the synthesis of procollagen III followed a delayed kinetics of alterations in hydrogel-treated vs. untreated burnt explants. Such a delay may be estimated to reach 2 days and was responsible for obvious differences in terms of procollagen III staining intensities on day 10 post burn when comparing hydrogel-treated vs. untreated burnt explants. On day 14, however, the same amount of procollagen III was observed in the center of the burnt area irrespective of the application of a poloxamer hydrogel.

The poloxamer gel caused a delay in re-epithelialization, with a delay in keratinocyte proliferation and a delay in the production of procollagen III. However, the gel did not prevent re-epithelialization and complete healing of both the epidermis and the dermis to occur on day 14.

## 3. Discussion

Until now, only few published studies reported on the use of human skin explant to perform ex vivo models of deep second-degree burns [12,16,18,23]. The experimental protocol we elaborated differs from those previously described with regard to at least two parameters discussed below: (i) the larger size of skin explants, and (ii) the application of a tension force. Our model was indeed carried out on large skin explants of 16 cm^2^ compared to an average of 1–2 cm^2^ in similar models [12,16,23]. Such a larger size is an advantage as it allows comparisons between different experimental conditions (e.g., treated vs. control) skin explants deriving from the same donor and cultured under the same conditions. Culturing the explant at the air-liquid interface, a condition rarely achieved in burn models performed on skin explants [12], is also critical in order to reproduce the native conditions of the human skin. Finally, the application of a tension force on skin explants is also potentially important as it was proposed to condition the extent of dermal tissue remodeling in experimental models of burns [12]. Indeed, skin fibroblasts are endowed with biomechanical properties allowing to sense physical forces applied to the dermis to translate these into an adapted molecular behavior, notably with regard to the synthesis of extracellular matrix (ECM) proteins and the acquisition of an α-SMA^+^ myofibroblast phenotype. While we did not observe any increase in the density of α-SMA^+^ myofibroblasts during the course of our model, we demonstrated a decreased synthesis of procollagen III in the burnt dermis, which, to our knowledge, has not been reported in previous models of experimental burns performed on human skin explants. Such a decreased expression observed on day 8 was followed by a return to baseline levels on day 14, while the expression pattern of collagen I and III remained unchanged during the same period of time. It should be noted that, although burns alter the molecular composition of the dermal extracellular matrix, immune cells (e.g., macrophages) are required to eliminate deposits of altered ECM proteins [24]. In our model, the absence of blood circulation, which prevents a significant recruitment of immune cells in the injured area, may explain why collagen I and III immunostaining appeared unchanged. In contrast, the decreased intensity in procollagen III staining reflects an altered secretory behavior of fibroblasts, which is not linked to the scavenging of ECM proteins by macrophages.

Interestingly, the re-epithelialization process in our model appeared to follow a similar kinetics of events to that observed in vivo in acutely injured human skins [25]. Such events notably include keratinocyte proliferation occurring initially at the wound edges and allowing the formation of an epidermal tongue which progressively covers the entire lesioned area in approximately 8–9 days. Moreover, as reported in vivo [5], the regeneration of the dermo-epidermal junction follows the same kinetics of progression than the tongue of epidermal regeneration.

An interesting finding of our study was that keratinocyte stem cells and progenitors robustly proliferated up to at least 14 days post burn. In contrast, keratinocyte proliferation was relatively low in control skin explants and tended to decrease over time in our culture conditions. This decrease in proliferative cells in the control skin could be due to the absence of friction. Indeed, it was previously shown that subtle alterations of the stratum corneum, similar to the physiological alterations induced in vivo by friction forces, are able to stimulate the proliferation of basal keratinocytes in 3D models of cultured skin cells [26]. The molecular mechanisms involved in the regulation of keratinocyte proliferation in 3D models of epidermal or whole-skin cultures are not fully understood. Trophic factors are possibly missing in our culture system. This holds true notably for blood-derived trophic factors such as transforming growth factor (TGF)-β1, epidermal growth factor (EGF), and platelet-derived growth factor (PDGF) secreted notably by the platelets during the healing process [27,28,29,30]. In the absence of fibroblasts, the weakening of the stratum corneum triggers a stimulation of cultured keratinocytes aimed at restoring the cutaneous barrier [26]. Thus, keratinocytes of cultured human skin explants maintain their proliferative potential; however, in the absence of a stimulus applied to the stratum corneum, they no longer proliferate. In any case, our results confirm previous studies pointing to the drawbacks of models using long-term ex vivo cultures of human skin explants [31,32]. It appears that maintaining such skin explants beyond 14 days of culture may induce experimental biases. This point is all the more important when studies rely on wound models requiring extended periods of culture. As compared to the excisional wound model described by Xu et al., we found that re-epithelialization following experimental burn on human skin explants is slower (8–9 days versus 6 days in the excisional wound model model). Nevertheless, our model still allows studying under unbiased conditions the early stages of burn healing, including re-epithelialization, EDJ repair, and procollagen III synthesis.

An important aim of our study was to determine whether our experimental model of second-degree burn was suitable for the therapeutic assessment of topically applied candidate molecules. A major issue in this context was first to validate the usability of a vehicle as a drug-delivery system to test healing drugs. Such a vehicle had to harbor at least two important features: (i) a gel-type texture allowing a prolonged contact of the tested therapeutic molecule with burnt skin area, and (ii) a previously demonstrated lack of local and general toxicity. We opted for a poloxamer p407/p188 hydrogel exhibiting several interesting physical properties. First, the poloxamer hydrogel p407/p188, as with other poloxamer hydrogel mixes, has the advantage of being thermoreversible. The hydrogel is a low-viscosity solution at room temperature and solidifies into a gel at temperatures >30 °C. This gelification is due to an aggregation of molecules in micelles [21,33]. This property makes the poloxamer hydrogel ideal for topical application as it may be easily handled in liquid solution and gels once applied to the skin, being thus maintained on the treated area. Currently, poloxamer hydrogel alone is not used in the treatment of burns. It is studied only as a vehicle for active ingredients. Of note, several studies demonstrated local tolerance to p407 poloxamer hydrogel when tested on the skin, including the burnt skin [22,34]. Moreover, a number of studies used a poloxamer hydrogel as a vehicle for bioactive molecules aimed at promoting skin healing [21,34,35,36,37,38]. These various studies highlighted the value of using a poloxamer hydrogel as a delivery system for an active ingredient. As an example, a poloxamer hydrogel allows the transcutaneous delivery of morphine, the analgesic effects of which were observed up to 24 h after application [35]. Furthermore, a hyaluronic-acid- and chitosan-containing p407 poloxamer hydrogel was successfully used for the topical application of vitamins (A, D and E) in a mouse burn model [34]. However, it should be noted that, in the abovementioned works, the impact of poloxamer hydrogel alone was not assessed. Indeed, to our knowledge, only one study tested the potential toxicity of p407 poloxamer hydrogel on skin cells [38], more specifically, the fibroblastic murine cell line NIH 3T3. In this regard, we present the first in-depth analysis of the impact exerted by p407/p188 poloxamer hydrogel on the healing process of cultured human skin explants. To our surprise, we found that p407/p188 poloxamer hydrogel delayed by several days the re-epithelialization of the burnt area. Nevertheless, the re-epithelialization process was completed before day 14 demonstrating the usability of this hydrogel in our model of second-degree burn. How p407/p188 poloxamer hydrogel impacts re-epithelialization remains to be established. A possible explanation could be that the accessibility of oxygen to the superficial skin layers of skin explants is somehow altered by the hydrogel and cannot be balanced by blood-borne sources of oxygen. Another possibility could be that poloxamer hydrogel induces local modifications of the tension forces and/or other physical parameters which are instructing the proliferation of keratinocytes in human skin explants. In any case, we show that, when testing bioactive hydrogels in our model of burn injury, a control condition consisting of the application of the hydrogel alone is indispensable for a reliable interpretation of results. The major interest of poloxamer hydrogel is its thermosensitivity, which allows easy use with a simple solubility of the active principle in order to test its impact. Future studies will be aimed at testing candidate healing molecules in our model, notably new molecules harboring more efficient antimicrobial and trophic properties than silver sulfadiazine, the currently most frequently used topical treatment for second-degree burns [10,11]. Finally, it should be noticed that beyond the therapeutic evaluation of topically-applied molecules, our model may be used for the assessment of strategies based on the graft of cells (for e.g., mesenchymal stem cells or keratinocytes) [39,40] of tissue (for e.g., epidermal cell sheets or 3D reconstructed skin) [41,42,43].

## 4. Materials and Methods

### 4.1. Ethical Statement

Skin samples were anonymized, and informed consent was obtained in accordance with the ethical guidelines from Lyon University Hospital (Hospices Civils de Lyon) and the principles of the Declaration of Helsinki. All the samples used in this study belong to a collection of human skin samples declared to the French research ministry (Declaration no. DC-2008-162 delivered to the Bank of Tissues and Cells of the Hospices Civils de Lyon).

### 4.2. Skin Explant and Burn Procedure

All skin explants were obtained from residual tissues generated from elective abdominoplasties. The workflow of our experimental procedures is depicted in Figure 9, and a step-by-step description is provided below.

Subcutaneous fat was carefully removed with a sterilized scalpel. Skin samples were cleansed with 70% ethanol, washed two times for 15 min in phosphate-buffered solution (PBS), and then soaked in Dulbecco’s modified Eagle’s medium (DMEM, Invitrogen, Carlsbad, CA, USA) supplemented with antibiotics (100 mg/mL gentamicin (Panpharma, Fougères, France), 400 IU/mL penicillin (Panpharma), and 4 mg/mL amphotericin B (Panpharma)) overnight at 4 °C.Skin explants were cut into small pieces of hexagonal shapes with sides measuring 2.5 cm. A sterilized blotting paper pattern of the desired size was used to allow a regular and reproducible cut.Sutures were performed on each corner of the shape in order to mount skin explants on metal grids and to prevent the occurrence of retractions.Grids were placed in Petri dishes (100 × 20 mm) containing medium, and explants were cultured in the liquid–air interface.Burns were performed with a 5 mm diameter metal rod which was immersed in a bath of hot water heated to 100 °C. The heated metal rod was applied for 1, 5, or 10 s on skin explants. Several burns were performed on each explant so that comparisons between time points and/or culture conditions could be performed on skin explants derived from the same donor (Figure 10). The debridement of skin lesions was performed on day 1 post burn with a sterile compress.

Two different media were tested: (i) medium A is routinely used for the culture of a large range of human cells and is composed of DMEM supplemented with 10% fetal calf serum (FCS) and antibiotics (20 mg/mL gentamicin (Phanpharma), 100 IU/mL penicillin (Phanpharma), and 1 mg/mL amphotericin B (Phanpharma)); (ii) medium B, used otherwise in our laboratory for the in vitro 3D culture of reconstituted human skins, is composed of DMEM supplemented with 8 mg/mL bovine serum albumin (Sigma, St Quentin Fallavier, France), 0.12 IU/mL insulin, 0.4 μg/mL hydrocortisone, and antibiotics. In both cases, culture media were replaced every other day.

### 4.3. Hydrogel Treatment

A poloxamer hydrogel was used as to test the feasibility of performing topical pharmacological assays on our model. The gel generously provided by Dr. Salmon (EA4169 “Fundamental, Clinical and Therapeutic Aspects of Skin Barrier Function”, University of Lyon 1, Lyon, France) was composed of 25% p407 and p188 (5:1) poloxamers diluted in 1× PBS. With this composition of poloxamers, the generated hydrogel is liquid at temperatures <30 °C and acquires the physical features of a gel at temperatures >30 °C. Since skin explants were cultured at a temperature of 37 °C, the applied hydrogel was maintained on the site of tested skin area.

The hydrogel was applied daily and removed each time with a sterile compress after 6 h of application.

### 4.4. Histologal and Immunohistological Analysis of Skin Explants

The skin explants were harvested at different time points following culture initiation. Under experimental conditions of burn, the whole lesioned area was excised and cut in two pieces which were either embedded in optimal cutting temperature compound (OCT) and stored at −20 °C until use or immediately fixed with 4% formaldehyde before being embedded in paraffin.

Tissue blocks were cut into 5 µm thick slides. For histological analysis, paraffin-embedded formalin-fixed samples, after dewaxing and rehydration, were stained with hematoxylin phloxine saffron (HPS).

For immunochemistry, tissue sections were incubated in 5% H_2_O_2_/3% normal goat serum (NGS; Jackson Immunoresearch, Suffolk, UK) to inactivate endogenous peroxidases. Nonspecific binding was blocked in PBS containing 4% bovine serum albumin (BSA; Sigma, St Quentin Fallavier, France) and 5% NGS. Slides were then incubated with primary antibodies overnight at 4 °C: collagen VII (dilution 1:50, clone 4D2, sc-33710, Santa Cruz Biotechnology, Inc., Hiedelberg, Germany) and HLA-DR/DP/DQ/DX (dilution 1:1500, clone CR3/43, sc-53302, Santa Cruz Biotechnology, Inc., Hiedelberg, Germany). Secondary horseradish peroxidase (HRP)-conjugated anti-mouse (Dako) was incubated for 1 h at room temperature. Labeling was revealed using 3,3’-diaminobenzidine (DAB; Dako) and slides were counterstained using Harris’ hematoxylin (25%, Sigma-Aldrich, St Louis, MO, USA). For collagen VII immunostaining, an initial step of heat-mediated and enzymatic antigen retrieval using trypsin (10 min at 37 °C, ab970, Abcam, Cambridge, MA, USA) was performed prior to incubation with the primary antibody.

For immunofluorescence, labeling was performed either on 5 µm thick paraffin-embedded formalin-fixed samples (for Ki67, procollagen III, and αSMA) or on air-dried 5 µm thick cryosections (for collagen I and III immunostaining). Slides were incubated overnight at 4 °C with the following primary antibodies: Ki67 (dilution 1:50, clone MIB-1, GA626, DakoCytomation, Glostrup, Denmark), procollagen III (dilution 1:500, clone M-58, Merck, Darmstadt, Germany), αSMA (dilution 1:200, NCL-L-SMA, Novocastra Laboratories, Newcastle upon Tyne, UK) collagen I (dilution 1:500, Novocastra Laboratories, Newcastle upon Tyne, UK), and collagen III (dilution 1:250, Novocastra Laboratories, Newcastle upon Tyne, UK). Sections were then incubated for 1 h at room temperature with secondary anti-mouse or anti-rabbit antibodies coupled to the fluorochrome Alexa 488 (Molecular Probes, Invitrogen, Carlsbad, CA, USA). Nuclei were visualized with Hoechst stain.

Image acquisition for Figure 1 was performed using an Eclipse 50i microscope (Nikon, Champigny sur Marne, France). All other images were acquired using a slide scanner (Axioscan Zeiss, Wetzlar, Germany) in order to enable a global view of the whole burnt area and of adjacent non-lesioned tissues.

### 4.5. Fluorescence-Based Detection of Intracellular H_2_S

Keratinocytes were cultured until sub-confluence on cover glasses in a 24-well plate. NaHS (0.25 or 2 mM) and H_2_S fluorescent probe (100 µM, P3, Sigma) were co-incubated for 1 h at 37 °C and 5% CO_2_. Cells were rinsed three times in PBS 1× and fixed in 4% formaldehyde. Cover glasses were then transferred on a slide. Detection of intracellular fluorescence was carried out on an Eclipse 50i microscope (Nikon, Champigny sur Marne, France).

### 4.6. Image Analysis

Image processing and analysis were performed using the Image J software (Research Service Branch, US National Institute of Health, Bethesda, MD, USA). Ki67-positive epidermal cells were counted and expressed as a percentage relative to the total number of nuclei in the epidermal area analyzed. For αSMA, collagen I, collagen III, and HLA, the surfaces covered by staining were measured and expressed as a percentage relative to the total dermal area analyzed. When needed, measurements were performed at both edges of the burnt area, as well as at the center of the lesioned area.

### 4.7. Statistical Analysis

For all data, the GraphPad Prism 4 software (GraphPad Software Inc., La Jolla, CA, USA) was used to determine statistical significance with the paired *t*-test; statistically significant differences are indicated by asterisks as follows: * *p* < 0.05, ** *p* < 0.01.

## Figures and Tables

**Figure 1 ijms-21-06956-f001:**
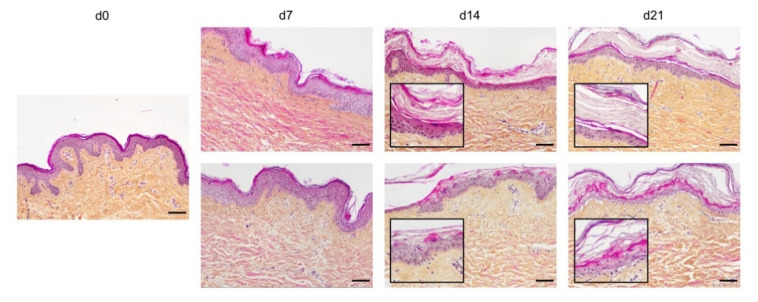
Assessment of culture media impact on ex vivo cultured human skin explants. Hematoxylin phloxine saffron (HPS) staining was performed on human skin explants harvested at different time points of the culture. Results obtained with two distinct culture media were compared: medium A (standard culture medium; upper panels) and medium B (medium used for three-dimensional (3D) cultures of skin cells; lower panels). From day 14 onward, skin explants cultured in medium B exhibited a detachment of the epidermis (bottom panels—days 14 and 21). Results obtained for each condition are representative of three series of experiments performed on skin explants derived from three donors. Scale bar: 100 µm.

**Figure 2 ijms-21-06956-f002:**
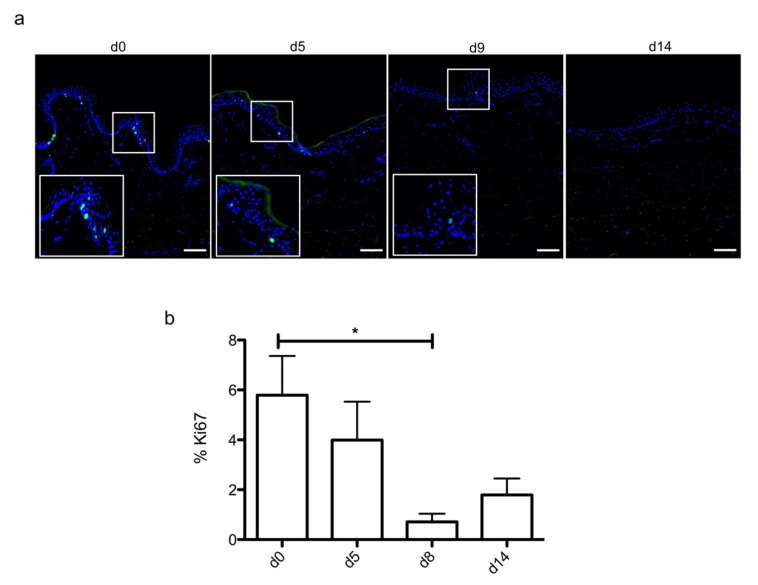
Demonstration of a persisting proliferating potential in the basal keratinocytes of cultured human skin explants. (**a**) Immunostaining of the proliferation marker Ki67 was performed on human skin explants harvested at different time points of the culture. Ki67^+^ keratinocytes were observed in the basal layer in freshly isolated skin explants (day 0) and were still detectable on day 5 following culture initiation. Only a few Ki67^+^ cells remained detectable in the basal layer of skin explants from day 9 onward. (**b**) The proliferation index of keratinocytes (ratio of proliferating cells relative to the total number of epidermal cells) was measured in human skin explants harvested at different time points of the culture. For each donor (*n* = 3) and for each time point, measurements were carried out on three distinct areas and expressed as mean values. Statistical significance was assessed with the paired *t*-test. *: *p* < 0.05. Scale bar: 100 µm.

**Figure 3 ijms-21-06956-f003:**
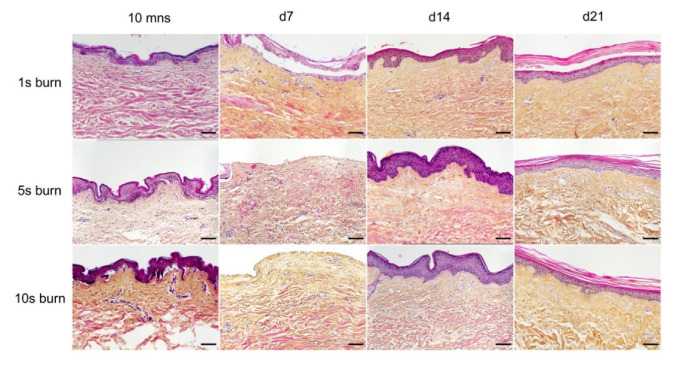
Assessment of burn duration impact on epidermal integrity in ex vivo cultured human skin explants. Hematoxylin phloxine saffron (HPS) staining was performed on human skin explants harvested at different time points post burn (10 min and days 7, 14, and 21). Burns were generated with a metal rod heated to 100 °C and applied for 1 s (upper panels), 5 s (middle panels) or 10 s (bottom panels). A 1 s burn impacted only the superficial layers of the epidermis, whereas 5 s and 10 s burns induced alterations of the whole epidermis. Under the three conditions tested, the burnt area was fully re-epithelialized on day 14. Images were taken in the center of the burn. Results obtained for each condition are representative of three series of experiments performed on skin explants derived from three donors. Scale bar: 100 µm.

**Figure 4 ijms-21-06956-f004:**
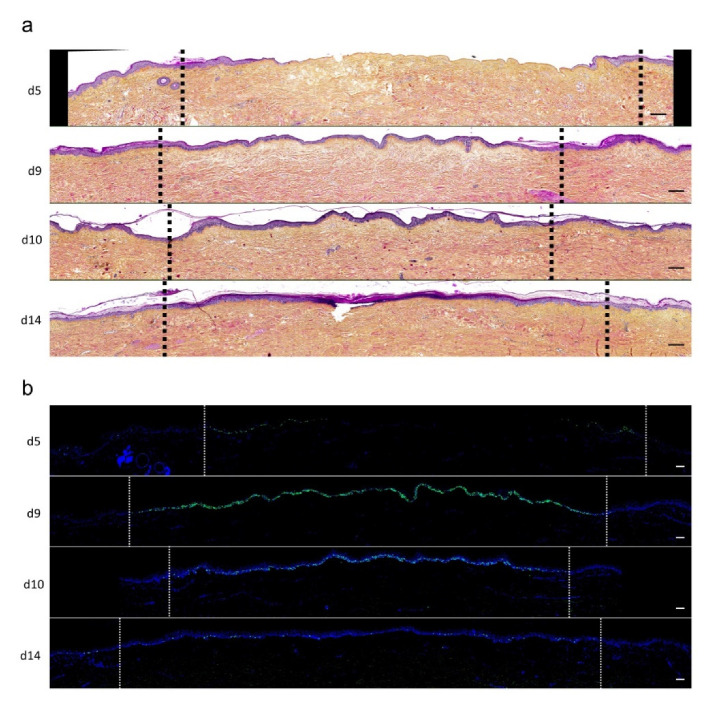
Kinetics of re-epithelialization from day 8 onward in ex vivo cultured human skin explants submitted to 10 s long experimental burn injury. Following a 10 s long experimental burn performed on ex vivo cultured human skin explants, the kinetics of re-epithelialization was assessed by HPS staining (**a**) and immunostaining of the proliferation marker Ki67 (**b**–**d**). (**a**) Wound closure was operated by proliferating keratinocytes which migrated from the wound edges and then formed an epidermal tongue on d5, before covering the whole lesioned area on day 8. A complete differentiation of the epidermis up to the formation of the superficial layers, including the stratum corneum, was then observed on day 14. (**b**,**c**) The keratinocyte proliferation index (ratio of proliferating Ki67^+^ cells relative to the total number of epidermal cells) was measured in human skin explants harvested at different time points of the culture. For each donor (*n* = 3) and for each time point, measurements were carried out on both edges of the lesion and expressed as mean values. On day 5 post burn, proliferating basal keratinocytes were only observed in the epidermal tongue, close to the wound margins. In this area, the keratinocyte proliferation index more than doubled as compared to the proliferation index observed in day 0 (control) skin explants (**c**). A peak in keratinocyte proliferation index was observed on day 8/9 with 55% of proliferating cells in the edges of the re-epithelialized lesion. Interestingly, several layers of Ki67^+^ epidermal cells were observed at this stage. Finally, on day 14 post burn, the keratinocyte index reached a mean of 10%, close to that observed in the epidermis of day 0 (control) skin explants (6%). (**d**) Demonstration of proliferating cells within the dermis in close vicinity to a hair follicle at the edge of the lesioned area (red arrows). Dashed lines in black (**a**) or white (**b**) delineate the burnt area from the adjacent non-lesioned skin areas. Statistical significance was assessed with the paired *t*-test. * *p* < 0.05, ** *p* < 0.01. Scale bar: 100 µm.

**Figure 5 ijms-21-06956-f005:**
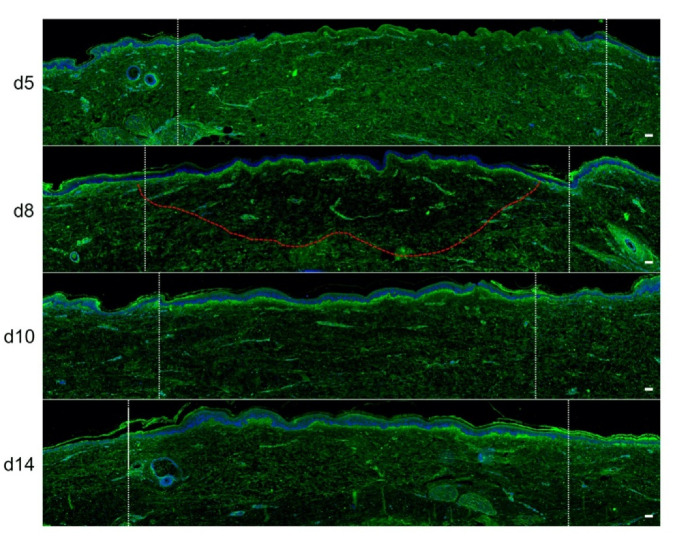
Kinetics of procollagen III synthesis in ex vivo cultured human skin explants submitted to 10 s long experimental injury. Immunostaining of procollagen III was performed on human skin explants harvested at different time points post burn (days 5, 8, 10, and 14). No significant change was observed on day 5. On day 8, a decreased intensity in procollagen III staining (area delimited by a red dotted line) was observed in the dermal area adjacent to the burnt epidermis. On day 10, this decrease in procollagen III staining intensity was less pronounced but still present until it returned to baseline levels on day 14. Experiments were performed on skin explants derived from three donors. White dashed lines delineate the burnt area. Scale bar: 100 µm.

**Figure 6 ijms-21-06956-f006:**
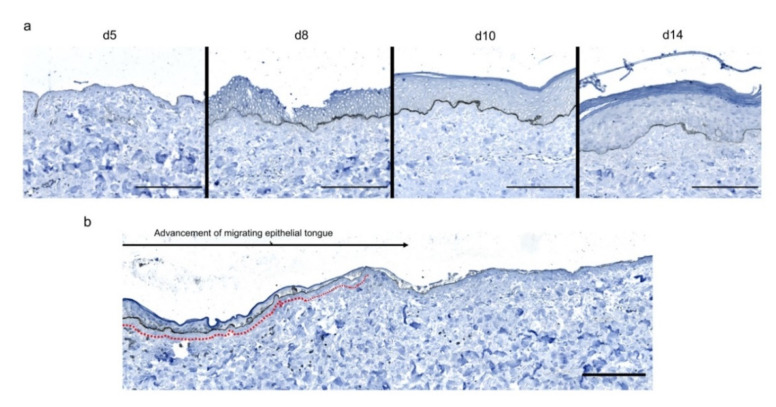
Kinetics of the dermo-epidermal junction (DEJ) repair in ex vivo cultured human skin explants submitted to 10 s long experimental burn injury. A 10 s burn was carried out on ex vivo cultured human skin explants and collagen VII immunostaining was performed on skin samples of the burn area harvested at different time points post burn (days 5, 8, 10, and 14. (**a**) In the center of the burnt area, no collagen VII remained detectable on day 5. However, in the same area, neosynthesized collagen VII was observed at the DEJ of the re-epithelialized epidermis on day 8 and onward. (**b**) In the wound margins, collagen VII staining was observed at the DEJ of the epidermal tongue formed by a few layers of cells. Experiments were performed on skin explants derived from three donors. Scale bar: 100 µm.

**Figure 7 ijms-21-06956-f007:**
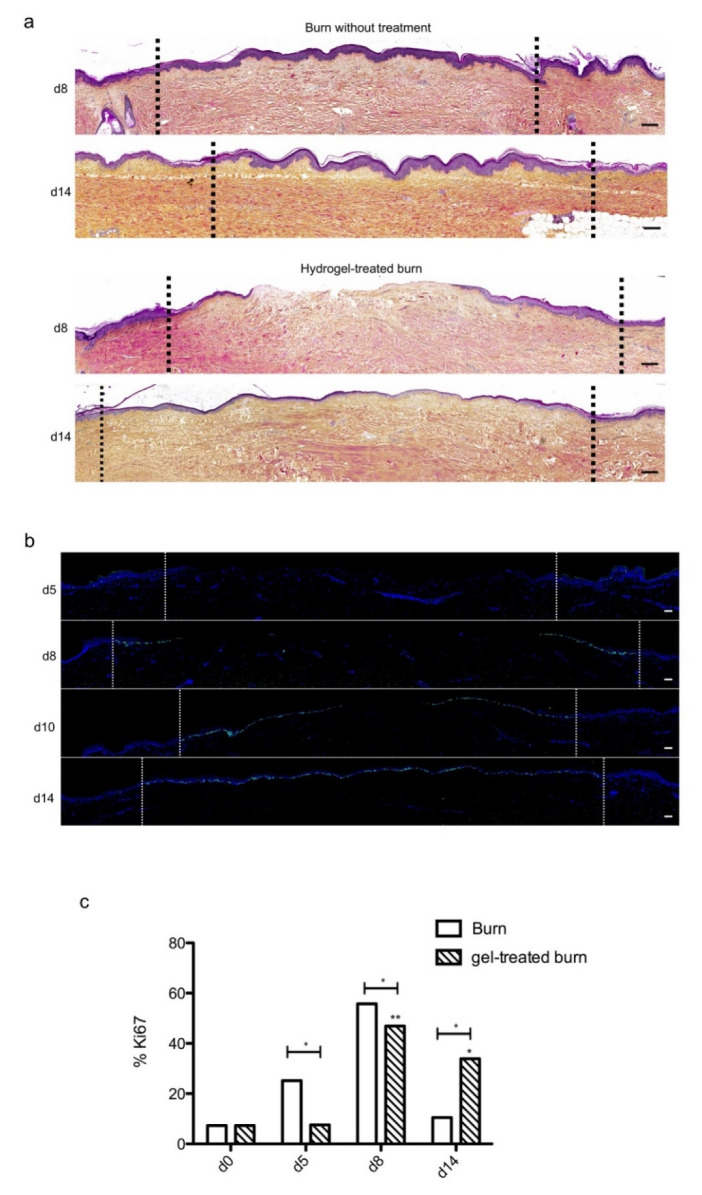
Kinetics of re-epithelialization in poloxamer hydrogel-treated ex vivo cultured human skin explants submitted to 10 s long experimental burn injury. A 10 s burn was carried out on ex vivo cultured human skin explants and, after wound debridement, on day 1, daily treatment with poloxamer p407/p188 hydrogel was applied. The impact of hydrogel applications on the healing process was assessed by (**a**) morphological analysis using HPS staining and (**b**) immunostaining of the proliferation marker Ki67. (**a**,**b**) On day 8 post burn, the hydrogel-treated lesion was not re-epithelialized, in contrast to the untreated lesion. The treated wound was re-epithelialized on day 14 but only 3–4 layers of keratinocytes were observed in the central burnt area, indicating that re-epithelialization was delayed by hydrogel treatment. (**c**) The keratinocyte proliferation index (ratio of Ki67^+^ proliferating cells relative to the total number of epidermal cells) was measured in human skin explants harvested at different time points of the culture. For each donor (*n* = 3) and for each time point, measurements were carried out on both edges of the lesion and expressed as mean values. No increase in proliferation index was noticed on day 5 in gel-treated lesions in contrast to the untreated lesions. On day 8, the proliferation index was similar in both conditions (47% in gel-treated lesions vs. 55% in untreated lesions). On day 14, the proliferation index in gel-treated burns was higher as compared to untreated burns (33% in gel-treated lesions vs. 6% in untreated lesions). Dashed lines in black (**a**) or white (**b**) delineate the burnt area. Statistical significance was assessed with the paired *t*-test. * *p* < 0.05, ** *p* < 0.01. Scale bar: 100 µm.

**Figure 8 ijms-21-06956-f008:**
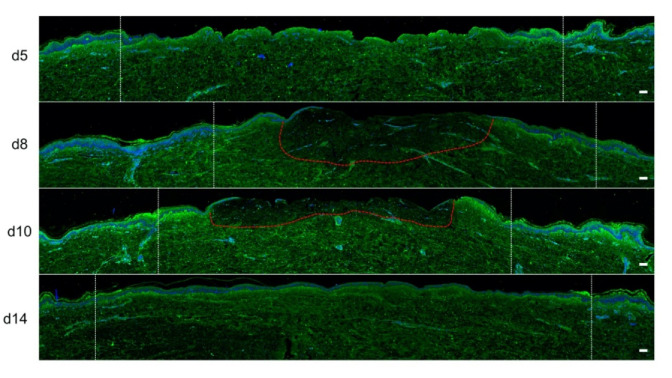
Kinetics of procollagen III synthesis in poloxamer hydrogel-treated ex vivo cultured human skin explants submitted to 10 s long experimental burn injury. Immunostaining of procollagen III was performed on gel-treated human skin explants harvested at different time points post burn (days 5, 8, 10, and 14). After wound debridement, on day 1, daily treatment with poloxamer p407/p188 hydrogel was applied. No significant change was observed on day 5 in the burn area compared to healthy skin. On day 8, a decreased intensity in procollagen III staining (area delimited by a red dotted line) was observed in the dermal area adjacent to the burnt epidermis. On day 10, this decrease in procollagen III staining intensity was less pronounced but still present until it returned to baseline levels on day 14. Experiments were performed on skin explants derived from three donors. White dashed lines delineate the burnt area. Scale bar: 100 µm.)

**Figure 9 ijms-21-06956-f009:**
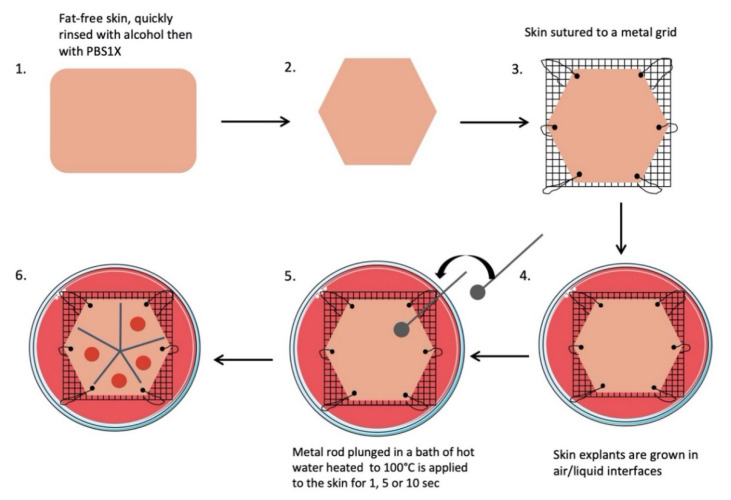
Scheme of the burn procedure on human skin explant model.

**Figure 10 ijms-21-06956-f010:**
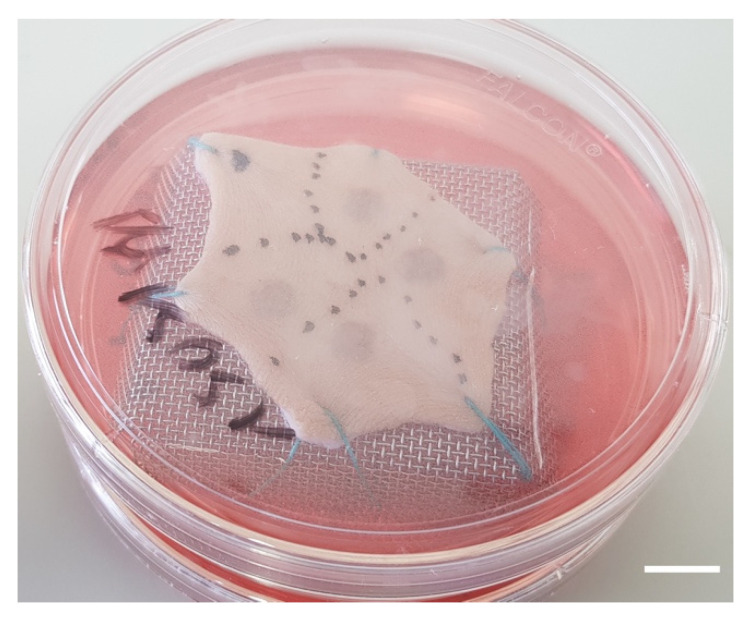
Picture of a human skin explant 12 h after experimental burn, before skin debridement (scale bar: 1 cm).

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
