# Peer review of "Characterization of a Topically Testable Model of Burn Injury on Human Skin Explants"

_ijms, 2020, doi:10.3390/ijms21186956_

Round 1

Reviewer 1 Report

The authors tested skin models of thermal injury.

The study is interesting, well designed, and supported by good graphic documentation. Interestingly, the authors found that hydrogel delayed thermal wound healing. It is quite surprising, as such hydrogels are frequently used to treat various wound types. The authors provided some hypothetical explanation reffering to a poor oxygen supply, however, in my opinion hydrogel might also prevent the diffusion of various growth factors from the medium to the wound. In that sense, the tested model does not reflect ideally the native conditions, as usually the epidermis is exposed to air, and not to aqueous conditions. Hydrogels are used to protect the epidermis and to prevent it from evaporation. It would be interesting to test, whether hydrogel would also delay reepithelization in the culture medium without any growth factors. Anyway, the authors should also discuss other potential explanations of delayed healing with hydrogel in the discussion. 

Reviewer 2 Report

The authors describe an original ex-vivo human skin model, which can be burned locally, and on which the cellular healing process can be measured. The model is suitable for testing of the therapeutic properties of various compounds, and could therefore help formalize research for skin healing e.g. in burnt patients.

The manuscript is well written and easy to read. Figures are clear. The methodology is adequately described.

Below are listed some comments which could help make the manuscript clearer:

P2L50 “ref 10 should be removed” as the reported technique is not enough scientifically validated

P2L71 “d0, d7, d14 and d21”: d for day ? same remark ; please amend

P2L72 “medium A (a standard culture medium) vs medium B (a culture 72 medium routinely used for 3D cultures of human skin cells)” : please explain the content of medium A and medium B ; if this is too long, it could be added as an appendix.

P2L81 “Ki67 staining of proliferative cells  was performed” ; please explain Ki67

P2L86 : “but this % drops” : please do not use symbols in text sentences ; please amend.

p4L126 “to perform further assess our model” unclear ; “to further assess our model” ?

p5L155 “the % of proliferating” : please do not use symbols in text sentences ; please amend

p7L192 “dermal remodeling in in our” : there is a repetition of “in”; please amend

p8L211 “the r the burnt” : unclear;

p9L248 “hydrogen” > hydrogel

p10L279 “the gel does not prevent re-epithelialization and complete healing of both the epidermis and dermis occurs at d14” ; grammar ; the gel does not prevent […] to occur at d14 ; please amend

p12L310 “A No significant change is observed” unclear ; please amend

p12L312 “adjacent to the r the burnt” unclear ; please amend

p14L377 “local and general toxicity We opted” ; there is a point missing; please amend.

p14L382 “as it may be be easily” : repetition of “be”

p14L383 “once applied to the skin is be maintained” ; unclear;

p14L384 “tested on theskin” > “the skin”

p14L389 “(Heilmann et al., 2013).” > reference to bibliography should be homogenous through the manuscript; please amend

p14L391 “(Soriano-Ruiz et al., 2020).”: same remark

p14L420 “residual tissues generated during came from elective abdominoplasties” : sentence unclear; there seems to be missing a word; please amend

p15L425: “washed 2 in phosphate buffer” unclear;

p15L425: “during 30 mns” : please the international abbreviations system ; please amend

p16L470: “applied hydrogen  > hydrogel
